# Real-Time Quality Index to Control Data Loss in Real-Life Cardiac Monitoring Applications

**DOI:** 10.3390/s21165357

**Published:** 2021-08-09

**Authors:** Gaël Vila, Christelle Godin, Sylvie Charbonnier, Aurélie Campagne

**Affiliations:** 1Univ. Grenoble Alpes, CEA, Leti, F-38000 Grenoble, France; gael.vila@univ-grenoble-alpes.fr; 2Gipsa-Lab, Univ. Grenoble Alpes & CNRS, F-38402 Grenoble, France; sylvie.charbonnier@gipsa-lab.grenoble-inp.fr; 3LPNC UMR 5105, Univ. Grenoble Alpes & CNRS, F-38040 Grenoble, France; aurelie.campagne@univ-grenoble-alpes.fr

**Keywords:** wearable cardiac sensors, electrocardiography, photoplethysmography, heart rate variability, signal quality, real-life measurements

## Abstract

Wearable cardiac sensors pave the way for advanced cardiac monitoring applications based on heart rate variability (HRV). In real-life settings, heart rate (HR) measurements are subject to motion artifacts that may lead to frequent data loss (missing samples in the HR signal), especially for commercial devices based on photoplethysmography (PPG). The current study had two main goals: (i) to provide a white-box quality index that estimates the amount of missing samples in any piece of HR signal; and (ii) to quantify the impact of data loss on feature extraction in a PPG-based HR signal. This was done by comparing real-life recordings from commercial sensors featuring both PPG (Empatica E4) and ECG (Zephyr BioHarness 3). After an outlier rejection process, our quality index was used to isolate portions of ECG-based HR signals that could be used as benchmark, to validate the output of Empatica E4 at the signal level and at the feature level. Our results showed high accuracy in estimating the mean HR (median error: 3.2%), poor accuracy for short-term HRV features (e.g., median error: 64% for high-frequency power), and mild accuracy for longer-term HRV features (e.g., median error: 25% for low-frequency power). These levels of errors could be reduced by using our quality index to identify time windows with few or no data loss (median errors: 0.0%, 27%, and 6.4% respectively, when no sample was missing). This quality index should be useful in future work to extract reliable cardiac features in real-life measurements, or to conduct a field validation study on wearable cardiac sensors.

## 1. Introduction

### 1.1. Broad Context

Among the host of wearable sensors implied in the *Quantified Self* nebula [1], physiological sensing stands out as a promising tool, as it may provide objective information on the subject’s inner states. Widespread applications are to be found in affective computing [2,3], an emergent and multidisciplinary research field that intends to assess and reproduce mental states from their peripheral and behavioral correlates. Wearable physiological sensors also pave the way for smart healthcare. Clinical issues, like chronic disease management or heart event prevention, may soon be achieved outside the hospital, thanks to advanced monitoring systems [4,5].

Whatever the end purpose, feature extraction algorithms (that compute certain parameters of interest) are usually run on physiological signals to detect or estimate the target state (e.g., subject’s stress level or risk for heart attack). In this process, cardiac activity through heart rate (HR) provides a major source of information. Advanced cardiac monitoring can be achieved through a set of statistical, frequency, or geometrical features, which quantify heart rate variability (HRV) [6]. Since they correlate with autonomic nervous activity, these features allow to predict target states or crisis events—such as stress [7], mental load [8], and epilepsy [9]. To reliably monitor these HRV features, the ideal cardiac sensor should be able to timely estimate the heart rate in ambulatory conditions and uncontrolled environments.

### 1.2. Wearable Technology for Heart Rate Estimation

The gold standard measure to extract HR is electrocardiography (ECG), in which heartbeats are markedly identified by sharp and prominent peaks (the R-waves). Instantaneous heart rates are derived from interbeat intervals (IBI), i.e., the time differences between successive heartbeats.

Wearable ECG can be achieved through textile electrodes (textrodes, or e-textiles [10]), which have shown performances comparable with the gold-standard Holter technology for heart rate estimation [11]. While Holter-type devices require expert knowledge to set up and become obtrusive in case of physical activity, textrode-based devices have been successfully integrated in chest belts or usual garments (smart T-shirts for instance). This makes e-textiles suitable for the widespread market and paves the way for real-life cardiac monitoring through ECG. To this day, however, they have been mainly commercialized in specialized areas, like exercise monitoring, and have not been embraced by the common people yet.

Meanwhile, wrist-worn sensors with abilities in cardiac monitoring (e.g., smart watches or smart wristbands) are now being used by an increasing numbers of users, worldwide and daily. Most of these sensors enable photoplethysmography (PPG), a standard measure of blood volume changes in the skin’s blood vessels [12]. PPG provides an oscillatory signal made of pulse waves, which can be used to identify heartbeats and estimate the IBI.

In controlled environments, the IBI signals extracted from PPG and ECG correlate well under static conditions [13,14]. However, HRV parameters have also shown substantial deviations between PPG and ECG under the influence of moderate mental effort or wrist activity [14]. Compared to the sharp R-waves in ECG, the oscillatory nature of PPG waveforms makes heartbeats difficult to locate in time with high accuracy. Moreover, the varying pulse transit time between the heart and the wrist may lead to a different estimate of the IBI at these two locations, depending on dynamic physiological factors like blood pressure [15]. To account for such differences, HRV is sometimes called pulse rate variability, when measured through PPG.

In ambulatory settings, an additional source of error in IBI estimation arises from motion artifacts, which especially affects PPG measurements. Over the past years, artifact identification and removal has been an active field of research on cardiac signals from both ECG and PPG [16,17]. A more comprehensive approach consists of computing a signal quality index (SQI) on the cardiac recordings. This index can be used to timely skip heartbeat detection when the raw signal is flawed.

### 1.3. Academic Research on Signal Quality

Regarding ECG, statistical or frequency parameters can be extracted from the raw signal to assess its quality or infer a signal-to-noise ratio [18]. These ECG features can be combined using machine learning techniques. Using support vector machines, the SQI presented in [19] obtained an accuracy of 93% in classifying 10 s segments of arrhythmic recording as clinically acceptable or not, with manual annotation as a reference. Regarding PPG, the pulsatile waveform can be compared to a template based on expert knowledge and the surrounding beats. This technique allowed [20] to reach 95% accuracy while predicting acceptable pulses against expert annotation, and [21] to reach a true positive rate of almost 100% in beat detection by setting a threshold on their own SQI—at the price of missing one beat over 10.

An alternative approach was presented in [22], where HRV was directly used to compute an SQI over 30 s segments of IBI signals from ECG recordings in ambulatory settings. Wavelet entropy was extracted from high-frequency ranges and fed in support vector machines, with expert annotation as the desired output. The final algorithm reached 94% in accuracy on the test set. An extension of this approach is proposed in [23], where the SQI (developed on both ECG and PPG recordings) was based on a mixed use of the IBI estimate (acceptable range and variations) and the raw sensor data (template matching strategy).

In each experiment, however, the data were collected in laboratory or clinical settings, using standard instrumentation (like Holter monitors or finger-worn PPG devices), presumably set up by some expert personnel. The proposed SQIs still need validation in real-life settings, where signals are much more exposed to motion artifacts and collected from commercial equipment installed by the users themselves. The quality index presented in this article was not designed to outreach these SQIs in accuracy for detecting flawed signal segments on standardized data. It aims to provide an easily understandable criterion to select reliable time widows in real-life IBI signals, assuming that beat detection is skipped in case of movement artifacts.

### 1.4. Advanced Cardiac Monitoring with Commercial Sensors

Indeed, some commercial devices compute their own SQIs to control the reliability of their heart rate outputs in case of noisy data. This is the case, for example, with the chest belt Zephyr BioHarness 3 that produces a confidence level on its ECG-based HR indicator (see below). Another well-known device in the sensor market is the smart wristband Empatica E4, based on PPG. To timely account for data quality, the device returns an IBI signal only when PPG waveforms are considered consistent enough by the heartbeat detection algorithm [24]. In other words, the IBI output is bound to some implicit, binary SQI whose computation has not been disclosed yet by the company.

The E4 wristband has already undergone several validation studies, which focused on the signal level (accuracy of IBI estimation) and/or the feature level (accuracy of the feature extraction process). For example, [25] estimated the number of missing beats in the IBI signal, relatively to standard ECG laboratory instrumentation; and compared statistical and frequency features from both devices. In the same vein, [26] compared IBI values and cardiac features from E4 with an ambulatory ECG device (VU–AMS) in clinical settings; and [27] collected cardiac features from E4 and another wearable ECG instrument (MindWare Mobile Device).

These studies are consistent in their findings. At the signal level, the IBI estimate from E4 is well-correlated with the IBI from ECG, with better results in resting than active conditions. In a given time window, however, the proportion of missing samples could reach 57% at rest and 99% during a talk [25], due to the heartbeat selection algorithm embedded in E4. At the feature level, the mean heart rate over a given time window is estimated with high accuracy; however, all features reflecting HRV show significant correlations [27] together with significant differences [26] with the ECG-based data.

This deviation from the gold standard reflects the well-known limitations of PPG in assessing HRV. That said, it also comes along with considerable loss of data on time intervals where the PPG is likely to be flawed by motion artifacts. Therefore, it is still unclear whether this substandard cardiac monitoring with Empatica E4 comes (i) from permanent limitations of PPG, or (ii) from transient data loss due to motion artifacts.

The impact of data loss on feature extraction (assumption (ii)) was shown in a simulation study [28], in which samples were removed from time windows of an ECG-based IBI signal, following a Gilbert burst model. As the missing sample rate increased, low-frequency components of the heart rate signal tended to be under-estimated, while the high-frequency components tended to be over-estimated. If data loss is actually a main source of error in the feature extraction process, there is room for a real-time quality management strategy by skipping feature computation when too many samples are missing. Still, this has to be shown in PPG-based heart rate data.

Finally, none of these studies has been conducted in real-life settings where motion artifacts are frequent and commercial sensors are set up by users themselves. In such conditions, a methodological obstacle has to be overcome: there is no gold-standard measurement to compare with the validated device. Since any wearable sensor (including ECG) is exposed to motion artifacts, there is a need to select reliable segments of IBI data to act as a benchmark. In that perspective, expert annotation cannot make a sustainable strategy; and academic SQIs still need validation on real-life data. Since most commercial SQIs are black boxes, they provide both help and burden to the researcher when it comes to selecting a commercial device for advanced cardiac monitoring. Our work attempted to address this methodological issue in a field validation procedure for sensor Empatica E4.

### 1.5. Outline of the Current Study

This article proposes the following contributions to current research on real-life cardiac monitoring systems.
A white-box SQI (the Lack Index) that quantifies the data loss in any IBI signal, and a straightforward criterion to select reliable IBI segments in real-world data,Validation results for a wrist-worn sensor (Empatica E4) on the field,Improvement reports when accounting for data loss in the feature extraction process.

In a first step, our index of data loss is developed using three properties of the IBI signal: its acceptable range, its acceptable variability, and the sum of its acceptable sample values. This SQI was used to identify time intervals where a wearable ECG device (Zephyr BioHarness 3) could be used as reference for heart rate estimation. In a second step, validity of sensor Empatica E4, a surrogate of the wearable ECG, is addressed at the signal level and at the feature level, in real-life data acquired by non-expert users. The validation method is similar to [25]: at the signal level, a beat-to-beat analysis is run to compare the IBI signals from both sensors. At the feature level, statistical and frequency parameters are compared between both signals. In this process, the Lack Index allowed us to select time windows in which large error rates were less likely to be encountered.

## 2. Materials and Methods

This study relies on a database acquired on several subjects, recorded during daytime for a whole working week. The experimental protocol and resulting dataset is introduced in the next paragraphs (Materials); then the main data processing and methods for elaborating the SQI is presented in detail (Methods), along with the validation procedure to estimate PPG data quality.

### 2.1. Materials: Experimental Protocol and Cardiac Sensors

#### 2.1.1. Recruitment Procedure

Three healthy male participants (aged 25, 27, and 33) agreed to wear a set of commercial sensors for a whole working week. None of them had a history of neurological disease, or followed any treatment susceptible to alter their cerebral or neurological functions. Before the experiment, each participant received a 2 h briefing, and a detailed manuscript explaining both the protocol and proper use of the sensors, which had to be taken home for the week. Sensors had to be equipped, taken off, and recharged everyday by the participants themselves according to the experimenter’s instructions, and then brought back to the lab for the final debriefing.

All participants signed a written consent at the end of the initial briefing, and received financial compensation after the equipment was returned. The experimental protocol was approved by the Ethics Committee in Non-Interventional Research (CERNI) related to COMUE Univ. Grenoble-Alpes (agreement N° 2015-05-12-67), and conducted in accordance with the Declaration of Helsinki.

#### 2.1.2. Wearable Sensors and Cardiovascular Signals

Two commercial wearable sensors were used to monitor cardiovascular activity: a chest belt and a smart wristband.

The chest belt Zehyr BioHarness 3 allows continuous electrocardiography (ECG_bh_, sampled at 250 Hz) by means of a couple of textrodes embedded on a chest strap. Acquisitions are recorded in a compact module clamped on the strap, which power supply and memory exceed a full recording day. The device automatically computes two kinds of heart rate estimators. The first one is a standard tachogram, sampled every time a new heartbeat is detected on signal ECG_bh_. The interbeat intervals (IBI_bh_) are computed by differentiating all timestamps. The second one is a custom heart rate approximation (HR_bh_), which is computed from the surrounding 15 s of signal IBI_bh_ and sampled every second. The reliability of this HR signal was asserted on the field in [29], in low- and high-physical activity conditions. As stated before, BioHarness 3 also returns its own (proprietary) SQI: for each sample HR_bh_, a confidence level (C_hr_) is returned as a percentage of maximum confidence (hence varying between 0% and 100%). According to BioHarness user manuals, this confidence level C_hr_ is computed from features expressing the maximum level of the raw signal ECG_bh_ and its signal-to-noise ratio. The device additionally provides: (i) 3-axis accelerometer data, which will be used below to compute the amount of body movement, and (ii) a respiration signal, which will not be considered in the current study.

The smart wristband Empatica E4 provides photoplethysmography with green and red light (PPG_e4_, sampled at 64 Hz) and automatically computes the interbeat interval (IBI_e4_). As mentioned earlier, signal IBI_e4_ is sampled when signal PPG_e4_ is considered reliable enough by the heartbeat detection algorithm. E4 additionally provides skin temperature and skin conductance data that will not be considered in the current study. It also provides 3-axis accelerometer data, which will be used below to estimate wrist activity. The device was worn on each subject’s non-dominant arm.

Both sensors can be used in local recording or Bluetooth transmission mode. The recording mode was preferred for battery saving and to avoid accidental connection losses. A main challenge with such a protocol choice is that both sensors had to be synchronized manually.

#### 2.1.3. Sensor Synchronization and Data Collection

As stated before, each sensor was set and started by participants themselves in the morning (i.e., as soon as possible after getting up). After self-calibration of the sensors (i.e., one minute after set up), the participants were instructed to stand still and perform three successive jumps at one-second intervals. This particular, contrived movement leaves sharp and salient marks on the acceleration signals of each sensor: this allows offline sensor synchronization with a time resolution high enough for our later analyses (around 0.1 s). The same procedure was applied before sensors unset at the end of the day (i.e., as late as possible before sleep), to account for time drift between both sensor clocks.

The participants’ activities and their interactions with the sensors are another paramount information when collecting data in real life during an extended period. The participants performed their daily routine wearing both sensors (office-like work, bike or car driving, leisure and sport training, two of them being high-level athletes). They were also asked during to report any meaningful daytime event by means of a short questionnaire filled on a homemade smartphone application. Moreover, they had to recall their main activities every evening by means of a spreadsheet questionnaire on the provided computer. The day reconstruction method followed the guidelines of [30]. After uploading sensor data and daily questionnaires to the laboratory’s own secured network, at the end of each day, they had a short interview with the experimenter by mail or by telephone, depending on participant’s own agenda.

#### 2.1.4. Preliminary Processing and Data Selection

Offline, all signal processing was performed using MathWorks^®^ Matlab R2017b. Sensor data were synchronized on the basis of the successive jumps (by identifying them and expanding signal timeframes for a perfect match), and examined in the light of our knowledge concerning subject activities during the experiments. Concerning heart rate data, the beat-to-beat analyses required perfect sensor synchronization. Therefore, every record in which the three successive jumps were not clearly identifiable on both sensors’ acceleration data, twice a day (since participants might have neglected them at one or the other end of the day), were discarded from current analysis.

Subsequently, heart rate signals from both sensors were superposed and visually examined. The HR estimator from BioHarness (HR_bh_) was inverted to match the values of the IBI estimator (IBI_bh_). It appeared that signal IBI_bh_ could not exceed 2^15^ milliseconds (32 s) in amplitude, suggesting that sensor BioHarness 3 stored its IBI values in a 16-bits variable. In our recordings, this had an adverse effect: as soon as no R-peak was detected during more than 32 s on the ECG, signal IBI_bh_ fell out of sync with signals HR_bh_ and IBI_e4_ (probably because of a memory overflow). This happened a couple of times on several recording days. To deal with this issue, IBI_bh_ was synchronized back with HR_bh_ manually and piecewise, by correlation maximization on each correct portion of signal IBI_bh_. Finally, the wristband signal IBI_e4_ was also synchronized with the chestbelt signal HR_bh_ (by correlation maximization) to account for the mean pulse transit time between the heart and the wrist.

During this two-step, high-precision synchronization phase, every recording day in which the proposed method needed further improvements was discarded from the final dataset. This left 11 recording days, i.e., 124 h of multimodal recordings.

To validate the heartbeat detection algorithm of BioHarness (which provides IBI_bh_) with an academic reference, the Pan-Tompkins (1985) analysis was finally run on each ECG recording [31,32]. This provided a third IBI signal, hereafter named IBI_pt_. Again, this signal was synchronized with other IBI estimators by correlation maximization.

### 2.2. Methods: Signal Processing and Quality Estimation

Our final goal was to compare the PPG-based signal IBI_e4_ with the ECG-based signal IBI_bh_, on time intervals where signal IBI_bh_ could be used as a benchmark. This implied to find such time intervals given that, unlike IBI_e4_, signals IBI_bh_ and IBI_pt_ were not corrected for flawed heartbeat detection. To this end, a preliminary rejection of outlier samples was performed on each IBI estimator following an automatic process. In each cardiac signal without outliers, the proportion of missing samples was then estimated over 1 min time windows of data. This proportion of missing samples is the output of our new SQI. To isolate time windows where no sample was missing (i.e., where flawless heartbeat detection could be assumed), a criterion was applied on this SQI. On time windows where signal IBI_bh_ could be used as benchmark, the quality of signal IBI_e4_ was finally assessed at the signal level (by matching the heartbeats from both signals) and the feature level (by extracted cardiac features from both signals).

#### 2.2.1. Outlier Removal

The IBI signal has inner properties that can be used to assess data quality. These properties were used to reject outlier samples on each IBI signal by applying two successive criteria: a range criterion and a variation criterion. The range criterion consisted of rejecting every sample that expressed an instantaneous heart rate outside the range [30; 250] bpm (kept wide to account for sport training events). The variation criterion considered that the value of an IBI sample cannot deviate from the mean of its neighbors by more than 30%. The whole outlier rejection procedure followed the guidelines of [33,34]. Since real-life IBI signals may sometimes be very noisy, rejection from the variation criterion was performed in a recursive fashion. A 5 s moving average was computed at first by focusing on each sample of the IBI signal, successively. Each IBI value deviating from this moving average by more than 30% was temporarily regarded as an outlier. In the following iteration, the moving average was computed on the remaining non-outlier samples. All IBI samples deviating from this new moving average by more than 30% were regarded as the new outliers, and so on. The procedure was run until no more outliers were found, or when the whole process exceeded 20 iterations.

Outputs of this two-step outlier removal process are illustrated in Figure 1. While computing the last moving average over 5 s windows, the number of available samples could vary from 1 to 16, depending on the local heart rate and number of missing samples.

#### 2.2.2. The Lack Index

In Empatica E4, IBI samples are provided only when the PPG signal is considered good enough by the sensor’s own algorithm. This results in a scarcer (yet more trustful) signal than the standard tacogram provided by BioHarness. After the outlier rejection step, however, the IBI samples from BioHarness were also scarcer when ECG signal quality was low. If the heartbeat detection algorithm truly detects heartbeats (this hypothesis will be discussed later), it can be assumed that sample scarcity (after outlier rejection) reflects the quality of the original IBI signal.

A simple index of “sample scarcity” can be designed by comparing the practical number N of IBI samples actually found in a given time window, with a theoretical number N_t_ of IBI samples that should be found in a standard tacogram with no outlier rejection. Let a time window of duration W, ranging within [t = 0; t = W], in which N heart beats are detected at times {t_1_, t_2_, …, t_N_}. The corresponding interbeat intervals are computed as: IBI_n_ = t_n_ − t_n−1_. In a raw IBI signal (i.e., without outlier rejection), all samples add up to the total time duration between the first and the last heart beat used for IBI computation. This can be expressed as:(1)$$\overset{N}{\underset{n=1}{\sum }}{\mathrm{IBI}}_{n}=\overset{N}{\underset{n=1}{\sum }}\left(t_{n}-t_{n-1}\right)=t_{N}-t_{0}\approx W$$
where t_0_ is the latest beat detected before the beginning of current time window (t = 0).

The mean IBI (µ) in the current time window is computed by dividing this sum of IBIs by the number of samples (N) found in this time window. Consequently, this number of samples N can be recovered by dividing the sum of IBIs by their mean µ. Since this sum of IBIs is close to W, according to Equation (1), one can also compute a theoretical number of samples N_t_ that is close to N:(2)$$N=\frac{\sum_{n=1}^{N}{\mathrm{IBI}}_{n}}{\mu }\approx \frac{W}{\mu }=N_{t}$$

In other words, when there is no missing sample in the IBI signal, the actual number of samples (N) that can be counted in any time window can also be estimated with a theoretical number (N_t_) derived from the window size (W) and the mean IBI (µ). When samples are removed from the IBI signal, however, the actual number N should decrease while the theoretical number N_t_ should remain stable. From the relative difference between these two numbers, we designed an estimator of sample scarcity, hereafter named the Lack Index (L). As shown in Equation (3) below, the Lack Index L can also be seen as the relative error between the sum of IBI samples and the time window length:(3)L=Nt−NNt=Wμ−NWμ=W−∑n=1NIBInW

The Lack Index estimates the proportion of missing beats in a given time window. Combined with an outlier rejection process, it evaluates the quality of the original IBI signal. In this study, three Lack Indexes (L_e4_, L_bh_ and L_pt_) were respectively computed out of the three IBI estimators (IBI_e4_, IBI_bh_ and IBI_pt_), on non-overlapping successive time windows where W was equal to 60 s. Regarding signal IBI_e4_, the Lack Index was used to quantify the lack of samples in real-life settings; and its impact on signal quality at the feature level. Regarding all IBI signals, the Lack Index was used to isolate time windows which can be considered as flawless, i.e., without any missing beat after outlier removal. This could be done by setting a rigorous threshold on its value, as shown in the next paragraph.

#### 2.2.3. Criteria to Select Flawless Time Windows

By definition, a given time window of length W encompasses all heartbeat times {t_1_, t_2_, …, t_N_}, but not t_0_ (located before t = 0) and t_N+1_ (located after t = W). Therefore, we can set the following inequality:(4)$$\left(t_{N}-t_{1}\right)<W<\left(t_{N+1}-t_{0}\right)$$

When the heartbeat detection is flawless (i.e., no outlier is found in the current window), Equation (1) applies for the sum of IBI samples and our Lack Index L is framed by two critical values:(5)$$\frac{t_{0}-t_{1}}{W}<\frac{W-\sum_{n=1}^{N}{\mathrm{IBI}}_{n}}{W}<\frac{t_{N+1}-t_{N}}{W}\;\frac{-{\mathrm{IBI}}_{1}}{W}<L<\frac{{\mathrm{IBI}}_{N+1}}{W}$$

Consequently, if the Lack Index L is superior to IBI_N+1_/W, then the current time window contains flawed beat detections for the current IBI estimator. Since there is no confidence that IBI_N+1_ has been well estimated, the minimum IBI value found in the time window is a more secure alternative to prevent adjacent time windows from influencing each other. Therefore, flawless time windows (i.e., windows where the IBI signal has no missing sample) can be identified for each of the three IBI estimators (i.e., IBI_e4_, IBI_bh_, and IBI_pt_) when the following criterion is satisfied:(6)$$\begin{array}{c} W*L<\min\left(\mathrm{IBI}\right) \end{array}$$

Alternatively, BioHarness provides its own SQI: the confidence level C_hr_, extracted from online properties of the ECG, whose accuracy is not under scope in the current study. This confidence level can also be used to select reliable time windows of cardiac signal from the BioHarness sensor, since it guarantees that heartbeat detection is safe each time its value reaches 100%. Hence, one may identify a flawless window of cardiac signal when the minimum value of C_hr_ over the time window is equal to 100%. This additional criterion can be used to validate the previous one in our attempt to find flawless windows on signals IBI_bh_ and IBI_pt_:(7)$$\begin{array}{c} \min\left(C_{\mathrm{hr}}\right)=100\% \end{array}$$

In the second part of this study, time segments of signal IBI_bh_ that were identified as flawless according to Equation (6) were used as benchmarks to assess the quality of signal IBI_e4_. This was done at the signal level and at the feature level through two distinct procedures: a beat-to-beat comparison and feature extraction from both signals.

#### 2.2.4. Beat-to-Beat Comparison between IBI Signals from BioHarness and E4

When both signals are properly synchronized, each beat in IBI_e4_ can be identified to one beat in IBI_bh_. This was done by splitting the timeframe of signal IBI_bh_ at the middle between each couple of successive heartbeats, defining a set of time intervals surrounding each heartbeat (see Figure 2 below). Samples of signal IBI_e4_ were enumerated within each of those time intervals: an empty interval means one missed beat and an interval with more than one sample IBI_e4_ means (at least) one overdetected beat. In each time interval where no heartbeat was missing or overdetected, a pair of matching IBI samples was identified between the two signals. The absolute difference was computed between all matching pairs of IBI samples.

This beat-to-beat analysis was run for every time window where signal IBI_bh_ had no missing sample, according to the criterion provided in Equation (6). The following parameters were computed on each single time window: (i) the probability for a sample in signal IBI_bh_ to be missing in signal IBI_e4_ (p^miss^), (ii) the probability for a sample in IBI_e4_ to be an overdetection (p^over^), and (iii) the mean absolute difference in IBI value between pairs of matched samples in signals IBI_bh_ and IBI_e4_ (μ^diff^). These parameters allow to quantify three types of errors that can occur during a beat detection: a false negative, a false positive, and a true positive with a wrong value. Let us call N_bh_ the number of samples for IBI_bh_ in this time window, N_e4_ the number of samples for IBI_e4_, n^miss^ the number of missing samples in IBI_e4_, n^over^ the number of overdetections, and ℳ the set of matched samples between the two signals. The three previous error rates were computed this way:(8)$$p^{\mathrm{miss}}=\frac{n^{\mathrm{miss}}}{N_{\mathrm{bh}}},{\;p}^{\mathrm{over}}=\frac{n^{\mathrm{over}}}{N_{e4}},\;\mu^{\mathrm{diff}}=\frac{1}{\mathrm{card}\left(ℳ\right)}\underset{n\in ℳ}{\sum }\left|{\mathrm{IBI}}_{e4}\left(n\right)-{\mathrm{IBI}}_{\mathrm{bh}}\left(n\right)\right|$$

#### 2.2.5. Feature Extraction

In advanced cardiac monitoring applications, descriptive features are usually computed over a full time window. For example, the mean interbeat interval (µ) and their standard deviation (σ, also called SDNN in current literature) are frequently used in the assessment of mental states. Some features are especially designed to describe HRV: for example, the root mean square of successive differences in the IBI (rmssd), and the low- and high-frequency components in its power spectrum (lf and hf, respectively, corresponding to the normalized power in frequency ranges [0.04; 0.15] Hz and [0.15; 0.50] Hz.

These five cardiac features were extracted and compared between signals IBI_bh_ and IBI_e4_, over time intervals where signal IBI_bh_ showed no missing sample (i.e., satisfied the criterion set in Equation (6)). The two frequency features were computed by integrating a Lomb-Scargle periodogram [35], a frequency analysis technique adapted to non-evenly sampled signals. The time window length W = 60 s, was set to ensure adequate resolution in the lower frequencies.

To assess feature extraction from IBI_e4_ with IBI_bh_ as the benchmark, the absolute error rate (E) between both estimates was computed for each feature f, over each time window.
(9)$$\begin{array}{c} f\in \left\{\mu ,\;\sigma ,\;\mathrm{rmssd},\;\mathrm{lf},\;\mathrm{hf}\right\},{\;E}^{f}=\left|\frac{f_{e4}-f_{\mathrm{bh}}}{f_{\mathrm{bh}}}\right| \end{array}$$

This error rate was used to answer two additional questions: (i) What levels of error could be expected for a given cardiac feature when extracted from signal IBI_e4_? (ii) To what extent could the Lack Index (L_e4_) be used to reduce this error? Since the Lack Index L_e4_ estimates the proportion of missing samples in a time window, question (ii) tests the hypothesis that lack of samples is a significant factor in misestimating a feature from signal IBI_e4_. If so, one could set a threshold on L_e4_ to timely control the risk of error when extracting a cardiac feature from real-life IBI recordings.

In practice, however, samples could be extremely rare in signal IBI_e4_ when the heartbeat detection algorithm was unsuccessful over large time intervals. In such contexts, it may be meaningless to compute some of the previous features since the IBI segment contains too little information. Therefore, some conditions were set to compute a given feature (and the corresponding error rate) only when the number of samples N_e4_ in signal IBI_e4_ was sufficient over a 60 s time window:µ was computed when N_e4_ ≥ 1 sample;σ was computed when N_e4_ ≥ 2 samples;rmssd was computed when N_e4_ ≥ 2 successive samples;lf and hf were computed when N_e4_ ≥ 18 samples.

The minimum number of samples needed to estimate the frequency features depends on the range and resolution of the Lomb-Scargle periodogram, which is different for each window of unevenly sampled IBI signal [36]. To avoid setting rules that are outside the scope of this study, the minimum N_e4_ was thus derived from the Shannon-Nyquist theorem, assuming that power spectrum should spread (at least) beyond the critical frequency of 0.15 Hz (hence: N_e4_ ≥ 60_[s]_ ∗ 2 ∗ 0.15_[Hz]_ = 18).

#### 2.2.6. Activity Level Monitoring

Since previous error rates were computed under ambulatory conditions, the amount of body movement stands as a major feature to monitor in order to explain potentially low performances for each IBI estimator. Using data from its 3-axis accelerometer, the BioHarness sensor computes and returns an estimate of activity which, according to device documentation, is derived from the Euclidean norm on the 3 bandpass-filtered acceleration components. Although this parameter might accurately reflect the amount of chest movement, there is no equivalent indicator for the wrist-worn sensor E4. A common Activity level estimator was thus computed from the raw acceleration data of both sensors, to quantify body movements on both locations on comparable magnitude scales. Each component was first bandpass-filtered in the frequency range [0.1; 10] Hz, with a digital second-order Butterworth filter, to account for non-human artefacts and the low-frequency contributions of gravity. According to previous studies in human movement quantification [37,38], the sum of each component’s average signal magnitude area correlates well with energy expenditure and allows to distinguish between rest and active periods. This can be expressed as:(10)$$A=\underset{i\in 1\ldots 3}{\sum }A_{i},{\;A}_{i}=\frac{1}{W}\overset{W}{\underset{t=0}{\int }}\left|a_{i}\left(t\right)\right|\mathrm{dt}$$
where a_i_ is one of the three bandpass-filtered acceleration components, and [0, W] delimits a given time window. Activity A was thus computed for both sensors to provide one activity estimate for each body location: A_bh_ for the chest and A_e4_ for the wrist. Over the set of all available 60 s time windows in the database, our activity estimate A_bh_ got a Spearman correlation coefficient of 0.91 with the mean activity estimate from BioHarness.

## 3. Results

This section presents validation results on sensor Empatica E4 through the three stages of our validation method: (i) selection of time intervals with no missing sample; then (ii) validation of estimator IBI_e4_ against estimator IBI_bh_ at the signal level and (iii) at the feature level.

### 3.1. Time Windows with Flawless Heartbeat Detection

For each of the three IBI estimators (IBI_bh_, IBI_pt_, IBI_e4_), the criterion shown in Equation (6) was applied to isolate 60 s time windows where no sample was missing. The criterion based on BioHarness native SQI (Equation (7)) was also applied as a reference. Following each criterion, the size of the selected subset of time windows is displayed in Table 1 below, as a percentage of the initial database. It represents the probability of selecting a random time window if either the BioHarness SQI (C_bh_) or the Lack index (L_bh_, L_pt_, L_e4_) had been used to select flawless time windows.

Each subset of time windows also contained various amounts of chest movement (A_bh_) and wrist movement (A_e4_). Within each subset, the range of these indicators reflects the amount of body movement that can be monitored when the corresponding criterion is satisfied. The two last lines of Table 1 thus displays the maximum A_bh_ and maximum A_e4_ over the time windows selected following each criterion. A lower figure means that fewer degrees of physical activity could be monitored through the corresponding signal without losing IBI samples.

In Table 1 (line 2, columns 2, 3, and 4), the number of selected windows is similar for the three cardiac signals from BioHarness (HR_bh_, IBI_bh_, and IBI_pt_): about half the original database was selected according to each SQI (resp. C_hr_, L_bh_, and L_pt_). Three conclusions can be drawn from these three figures. First, no cardiac sensor is immune to real-life artefacts, since half the database showed at least one missing beat after outlier rejection. Second, BioHarness algorithm for heartbeat detection (column 3) performed slightly better than the academic reference (Pan-Tompkins, column 4) on the same ECG recordings (with resp. 48% and 43% selected time windows). Third, the two criteria used for columns 2 and 3 validated each other in assessing BioHarness cardiac signals. Indeed, the proportions of the selected time windows are close to each other (resp. 51.4% and 48.2%); and 91% of the time windows that satisfied Equation (6) (based on the Lack Index L_bh_) also satisfied Equation (7) (based on the Confidence Level C_hr_). The native BioHarness SQI thus validated the ability of the Lack Index to select IBI segments with good signal quality.

Compared to the three cardiac signals from BioHarness, however, our criterion selected a very small subset of time windows showing no data loss for signal IBI_e4_ (0.6% of the original database). This significant shrinkage of the subset size (line 2) comes with a significant drop in the amounts of body movement that can be monitored in this subset (lines 3 and 4). Indeed, the maximum levels of chest activity (A_bh_) and wrist activity (A_e4_) are identical across columns 2, 3, and 4 (for sensor BioHarness), up to 36% (for chest) and 50% (for wrist) of the maximum activity level in the original database. In column 5 (sensor E4), though, the maximum levels of chest and wrist activity are 20 times lower, down to 1.8% (for chest) and 2.3% (for wrist) of the maximum activity levels in the database.

In a nutshell, Table 1 shows that time windows where signal IBI_e4_ can be considered flawless are very rare in real-life recordings. As expected, the quality of data from sensor Empatica E4 is strongly related to the amount of body movement. The next paragraphs address this issue more accurately, by validating the IBI estimate of Empatica E4 at the signal and the feature level.

### 3.2. Characterization of Empatica’s IBI Estimate at the Signal Level

In the next step, all time windows where signal IBI_bh_ showed missing samples (according to Equation (6)) were removed from the original dataset. This left 3370 time windows (56 h of recordings) where signal IBI_bh_ could be used as benchmark to validate signal IBI_e4_. Following the heartbeat-to-heartbeat analysis proposed in Section 2.2, we used this new dataset to address the following questions:What kinds of errors can be expected in signal IBI_e4_ under real-life conditions (Section 3.2.1);To what extent are these errors related to wrist movements (Section 3.2.2).

We finally used our results to show that the Lack Index accurately estimates the proportion of missing beats over any time window (Section 3.2.3).

#### 3.2.1. Error Rates over All Time Windows

As introduced in Section 2.2, three parameters were computed to qualify the estimation of the IBI by Empatica E4: the proportion of missing samples p^miss^, the proportion of overdetected beats p^over^, and the mean absolute deviation between matching IBIs μ^diff^. The repartition of these error rates is pictured with histograms in Figure 3 below.

Results on the missed sample rate p^miss^ confirm that signal IBI_e4_ was very scarce over the time windows where signal IBI_bh_ could be used as benchmark. In plot (a), half the time windows have a p^miss^ over 98%. Actually, 42% of the time windows did not show any sample of signal IBI_e4_. In 90% of the time windows, the heartbeat detection algorithm missed at least one beat over two (i.e., the 1st decile in p^miss^ is close to 50%).

In plot (b), however, one may notice that 81% of the time windows show no ovedetected heartbeat (p^over^ = 0%). Overdetection is thus a marginal phenomenon in signal IBI_e4_. In plot (c), one may also notice that deviations between matched samples of signals IBI_bh_ and IBI_e4_ are typically low: 90% of the time windows show a mean deviation (µ^diff^) below 130ms, which represents roughly 15% of a typical IBI value.

These results show that in the estimation of IBI with Empatica E4, the risk of false positive (i.e., an overdetected beat) is very low and the estimation accuracy is typically high. That said, plot (c) also demonstrates that such an estimation is not perfect: the mean value of µ^diff^ over the dataset is 67 ms, which represents roughly 7% of a typical IBI. Together with the high rates of missed beats, this should impact the quality of feature extraction in advanced cardiac monitoring applications.

#### 3.2.2. Impact of Wrist Activity

One may still argue that histograms of Figure 3 include all amounts of body movements, while PPG measurements are known to behave well under conditions of low physical activity. In this paragraph, we measure the same error rates at low and high amounts of wrist movement.

To distinguish several amounts of wrist movement, the dataset was divided in 10 subsets of time windows corresponding to each decile of wrist activity A_e4_ (the lowest 10% make one subset, the next 10% make another subset, etc.). In each of these subsets, the median and interquartile range were computed for the two error rates p^miss^ and µ^diff^ (p^over^ was not considered since it was typically negligible). The results are displayed in Figure 4.

In chart (a), the median (thick line) and the interquartile range (colored area) of the proportion of missing beats (p^miss^) together show a steep rise across the lower deciles of wrist activity (A_e4_ < 17 mG), and remain stable afterwards (p^miss^ ≈ 100%). According to the Student’s two-sample *t*-test, there was a highly significant difference in p^miss^ between the two first deciles of wrist activity (*p* < 0.001, 672 degrees of freedom). These results confirm that wrist movements strongly increased the probability of missing a heartbeat, even at the lowest rates (median p^miss^ ≈ 80% in the second decile of A_e4_).

In chart (b), the mean absolute deviation (µ^diff^) shows a comparable trend: the median error increases across the first deciles of wrist activity (from 21 ms at 4 mG to 36 ms at 17 mG); and tends to remain stable afterwards. A one-way ANOVA across all deciles of A_e4_ showed a significant impact of this factor on parameter µ^diff^ (*p* < 0.001, 1952 degrees of freedom). However, the median curve remains encased in parameter variability (colored area); and there is no significant difference between the two first deciles of wrist activity (*p* = 0.283, 563 degrees of freedom). Compared to p^miss^, the mean deviation µ^diff^ was thus only mildly affected by the amount of wrist movement.

During the final step of this study (see Section 3.3), the first decile of wrist activity was used to delimit a subset of time windows showing better IBI estimation, since p^miss^ and µ^diff^ are both lower than in the rest of the dataset.

#### 3.2.3. Validation of the Lack Index

In the current study, the Lack Index (L_e4_) and parameter p^miss^ are two distinct measures of the proportion of missing beats in a given time window. Over all time windows where signal IBI_bh_ could be used as benchmark, the linear correlation coefficient found between L_e4_ and p^miss^ was 0.999. This suggests that, despites the underlying hypotheses, our Lack index precisely estimates the actual proportion of missing samples in a time window. The mean absolute deviation between the two indicators is 0.6%. Since an extremely low IBI (e.g., 0.5 s) covers 0.8% of a 60 s time window, the minimum IBI divided by window length (60 s) was a relevant upper limit on L_e4_ to claim that a time window had no missing sample (p^miss^ = 0%). This validates the criterion proposed in Equation (6).

### 3.3. Validation of Empatica E4 at the Feature Level

Finally, the reliability of signal IBI_e4_ for advanced cardiac monitoring was addressed by extracting the five features introduced in Section 2.2.5. For each feature f ∈ {μ, σ, rmssd, lf, hf}, the corresponding error rate E^f^ between signals IBI_e4_ and IBI_bh_ was computed following Equation (9).

These error rates were first computed over the 3370 time windows where signal IBI_bh_ could be used as benchmark (i.e., where the Lack index L_bh_ satisfied the criterion defined in Equation (6)). The results are displayed in Figure 5 below. In each of the five panels, the plain blue line materializes the cumulative probability of error for a given feature *f*. This function corresponds to the probability for error E^f^ to be less than or equal to a given value. The median error is obtained when the curve reaches 50% in ordinate. The full curve can be interpreted like a ROC curve: when it rises steeply from 0% to 100%, lower error rates come with higher probabilities, which means that feature extraction from signal IBI_e4_ is more reliable.

When features are blindly computed over all time windows (plain blue line) the median errors are, in ascendant order: µ: 3%, σ: 25%, lf: 25%, rmssd: 62%, hf: 63%. The mean IBI (µ) by far is the more accurately estimated: almost 90% of the time windows have an error rate below 10%. The curve of each other temporal feature resembles the curve of a frequency feature: σ and lf on the one hand, rmssd and hf on the other hand. Such similarities, which justified the positioning of the features on Figure 5, are not surprising since these couples of features are known to correlate each other in long recordings [39]. In panels (b) and (c), the long-term HRV features (σ, lf) show together lower error rates than the short-term HRV features (rmssd, hf), but also higher rates than the mean IBI (µ).

To reduce such error rates and ensure a reliable feature extraction, one may set a criterion to select time windows where signal IBI_e4_ should be more reliable. As stated in Section 3.2.2, for example, the first decile of wrist activity (A_e4_) defines a subset of time windows where signal IBI_e4_ shows lower error rates at the signal level. The cumulative probability of error for this “Low Wrist Activity” subset is materialized by the red dashed lines in all panels of Figure 5.

Since we have designed a SQI (the Lack Index) that estimates the proportion of missing samples in a time window, we also questioned the influence of this parameter on feature estimation quality. To mirror the previous segmentation over wrist activity (A_e4_), the first decile of the Lack Index (L_e4_) made another subset of time windows with “Low Lack index”. The cumulative probabilities of error in this subset are provided by the green, dash-dotted line in each chart of Figure 5.

Finally, a third subset of data was made from the time windows where signal IBI_e4_ had no missing sample, according to Equation (6). This more stringent criterion was designed to suppress the effect of data loss during feature extraction. In each chart of Figure 5, the cumulative probabilities of error in this subset are represented by the green dotted line.

As expected, the “Low Wrist Activity” data (red dashed lines) yielded lower error rates than the full dataset. The “Low Lack Index” subset, however, yields better results for every feature, except hf. Regarding signal quality for feature extraction, therefore, the Lack Index seems to have greater discriminative power than physical activity estimates.

Indeed, the “No Missing Sample” criterion (based on the Lack Index) performed by far the best segment selection. In chart (d), the green dotted curve is merged with the top of the graph: this means that all time windows showed error rates close to 0%. In the other charts, however, the feature estimation is still imperfect; the median error rates being: σ: 10%, lf: 6%, rmssd: 34%, hf: 27%. If the long-term HRV features (σ, lf) now display low error rates, this error remains high for the short-term HRV features (rmssd, hf), even when no sample is missing in the time window.

Beyond its impact on feature extraction, the low number of available samples in IBI_e4_ also reduced the number of time windows available for feature computation. Over all time windows where IBI_bh_ could be used as reference, the mean IBI (µ) could be extracted 58% of the time (since at least 1 sample was required); features σ and rmssd, 51% of the time (since 2 samples were required); and features lf and hf, 23% of the time (since 18 samples were required).

## 4. Discussion

As a reminder, the contributions of this study are threefold: (i) a validation experiment of Empatica E4 on the field; (ii) a white-box SQI (the Lack Index) and a methodological framework to qualify heart rate signals in real-life settings; (iii) the use of the Lack Index to reduce the impact of data loss during feature extraction. The following paragraphs will discuss these three contributions.

### 4.1. Field Validation of Empatica E4

Technically speaking, Empatica’s built-in algorithm (that returns an IBI estimation only under conditions of satisfactory PPG quality) is working quite well. In our dataset, overdetection of heartbeats remained a marginal phenomenon. The IBI estimation error was typically low and this misestimation degree increased only moderately with physical activity (Figure 4b). In other words, Empatica’s heartbeat detection algorithm is both specific (few false positives) and accurate (true IBI values). In time windows where all heartbeats were detected by Empatica’s algorithm, perfect computation of the mean IBI (Figure 5d) demonstrates that signal IBI_e4_ was not biased. These results confirm the potential of PPG measurements for monitoring instantaneous pulse rates, which is already intended by a number of wrist-worn cardiac sensors in commercial settings.

At the signal level, such an accuracy is achieved at the price of a very scarce IBI estimation, as shown by parameter p^miss^ in Figure 3a. This lack of available IBI samples in real-world data supports the findings of earlier laboratory studies [25]. At the feature level, our study demonstrated that data loss is a major source of error when trying to extract HR and HRV features: reducing the proportion of missing samples also reduces the error rates, as seen in every chart of Figure 5. This factor is bounded to (but not entirely explained by) the amount of wrist activity (see Figure 4a).

### 4.2. Assets and Drawbacks of the Lack Index to Assess the Quality of a Signal

To produce these results, we used a two-steps method to conduct a validation study in real-life settings: (i) select time intervals where a reference measurement (e.g., wearable ECG) shows no missing sample after outlier rejection; and then (ii) compare the target heart rate signal with the benchmark IBI segments. Our method relies on a new white-box SQI: the Lack Index, which precisely estimates the proportion of missing samples in the IBI signal (see Section 3.2.3). This SQI comes with a straightforward criterion to select time windows in which no sample is missing (Equation (6)).

That said, the Lack Index assumes that all IBI samples in a time window are valid heart rate data. To be used as an SQI, it should be combined with an efficient outlier rejection procedure to ensure that all IBI samples have the properties of a valid cardiac signal. The procedure implemented in Section 2.2 is a perfectible example of such an approach: since it relied on closely neighboring data, it was unable to identify a too long succession of invalid IBI samples. For such reasons, future work in this area should consider different time windows or more advanced criteria for outlier identification (e.g., adaptive filtering [40]), provided the algorithm can withstand the fast variations and high missing sample rates usually found in real-world heart rate data.

### 4.3. Potential of the Lack Index in Advanced Cardiac Monitoring Applications

Regarding Empatica E4, the results of Table 1 and Figure 3 suggested that real-life cardiac monitoring applications should always rely on incomplete segments of the IBI signal (no sample was missing in only 0.6% of the dataset; and half the samples were missing in 90% of the dataset). To implement a sustainable feature extraction process, a compromise should be brought on an “acceptable” number of missed beats for each individual feature. The Lack Index was designed to deal with such an issue.

In Figure 5, indeed, the flawless detection criterion (Equation (6)) could suppress the estimation error for the mean heart rate (µ). This was not the case, however, for the HRV features (σ, rmssd, lf and hf) that still showed non-zero error rates when no sample was missing. As stated earlier in this study, these residual errors may arise from two distinct phenomena: (i) the unavoidable noise in IBI estimation due to the smooth shape of the PPG waveform, or (ii) the varying pulse transit time between heart and the wrist. Since factor (i) varies from one heartbeat to another, it may explain the residual error rates found for short-term HRV features (rmssd, hf). Since factor (ii) depends on physiological variables like blood pressure, it may explain the residual error rates found for longer-term HRV features (σ, lf). Another striking result is that residual errors are typically lower for longer-term (σ, lf) than for shorter-term (rmssd, hf) HRV features. This finding is consistent with [28]; it shows that high-level applications (e.g., inner states monitoring or heart event prevention) using high-frequency HRV should carefully consider the missing sample rate when running its predictions.

In this study, we did not interpolate the IBI signal since we were interested in the impact of data loss on feature extraction. Based on the Lack Index, future work may consider the use of interpolation approaches at different missing sample rates on real-world measurements (similar work has been done with simulated data loss [41]). Another limitation of this study is that HRV features were computed without accounting for the influence of the window size (W = 60 s). Future work should also consider the combined effect of the Lack Index (L) and the window size (W) while extracting features from real-life recordings (similar work has been done with simulated data loss [28]). Bearing that in mind, whether or not pulse rate variability and HRV should be considered as distinct cardiac measures is still an open question. The answer might depend on the features of interest, the time intervals, and the desired application.

As illustrated in this article, there is still work to be done on cardiac monitoring systems (using either PPG or ECG) to make the estimation of heart rate more robust to real-life conditions. Cardiac measurement will likely remain prone to artefacts, in spite of the technical advances made in the recent years. This illustrates the need for system-specialized SQIs (like the one proposed by BioHarness) to ensure that higher-level algorithms will not run on false information. In advanced cardiac monitoring applications, however, there is also need for white-box indicators to implement strong quality management strategies in the feature extraction process. In that perspective, the current study proposed a method to timely control the risk of error due to data loss in real-life settings. At a time when cardiac monitoring techniques are being developed beyond ECG and PPG, this method should also help the researcher to characterize several sensors outputs comparatively.

## 5. Patents

The work reported in this manuscript has led to patent applications, currently registered under N° FR3098390-EP3763283-US2021007674.

## Figures and Tables

**Figure 1 sensors-21-05357-f001:**
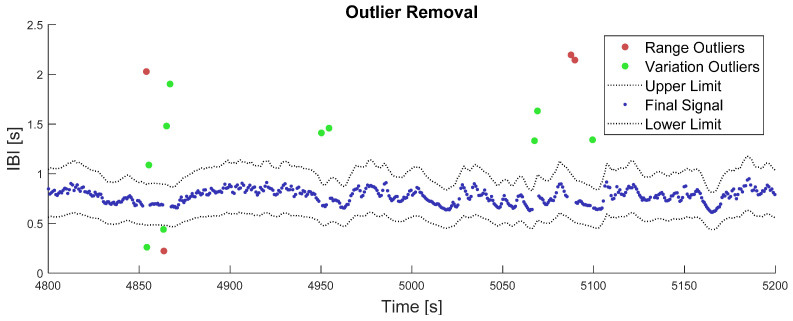
Illustration of the outlier removal process. The red dots represent outliers that were removed in the first place, from the range criterion on IBI values. The green dots represent outliers that have been removed in the second place, from the variation criterion on the IBI signal. The grey dotted-lines represent the final lower and upper limits after algorithm convergence on the variation criterion, i.e., 30% deviation from the moving average. The blue points represent the remaining IBI signal after outlier correction.

**Figure 2 sensors-21-05357-f002:**
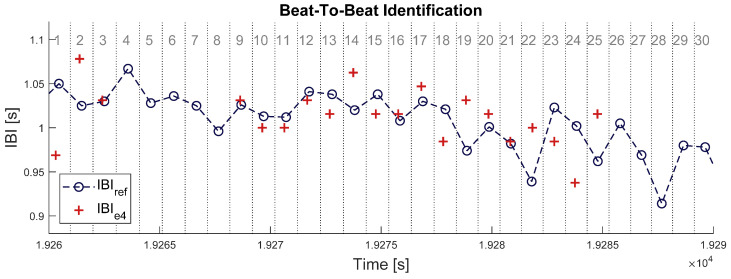
Illustration of beat-to-beat identification procedure in a 30 s time window. Blue circles with dashed line represent the IBI estimate from Zephyr BioHarness 3 (IBI_bh_). Red crosses represent the IBI estimate from Empatica E4 (IBI_e4_). Vertical dotted lines delimit the time intervals attached to each IBI sample from BioHarness, which are numbered in grey at the top of the chart. Intervals N° 4, 5, 6, 7, 8, 26, 27, 28, 29, and 30 have no red crosses; thus, 10 samples (33%) are missing in signal IBI_e4_. All red crosses here are matched samples, so that no sample is overdetected in signal IBI_e4_.

**Figure 3 sensors-21-05357-f003:**
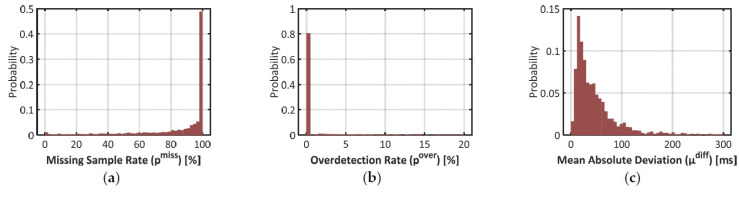
Histograms on three error rates to validate signal IBI_e4_ with signal IBI_bh_ as the benchmark. (**a**) Proportion of missing samples (p^miss^) in signal IBI_e4_; (**b**) Proportion of excess samples (p^over^)in signal IBI_e4_; (**c**) Mean Absolute Difference (µ^diff^) between matching pairs of samples between IBI_e4_ and IBI_bh_.

**Figure 4 sensors-21-05357-f004:**
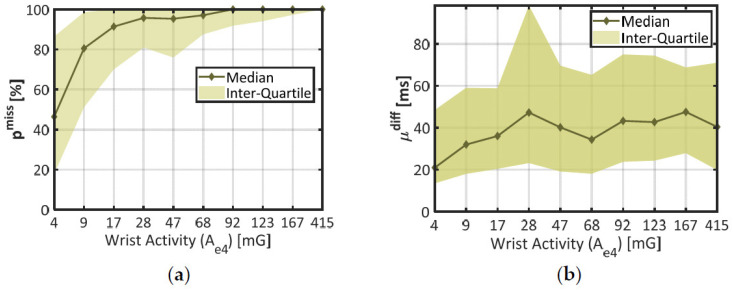
Statistics on IBI estimation quality through distinct levels of wrist activity. (**a**) Proportion of missed beats (p^miss^) against wrist activity (A_e4_); (**b**) Misestimation degree (µ^diff^) against wrist activity. The X-Axis represents each decile of A_e4_ in the dataset. Plain lines stand for the median error at each decile of activity, and colored areas materialize the interquartile range.

**Figure 5 sensors-21-05357-f005:**
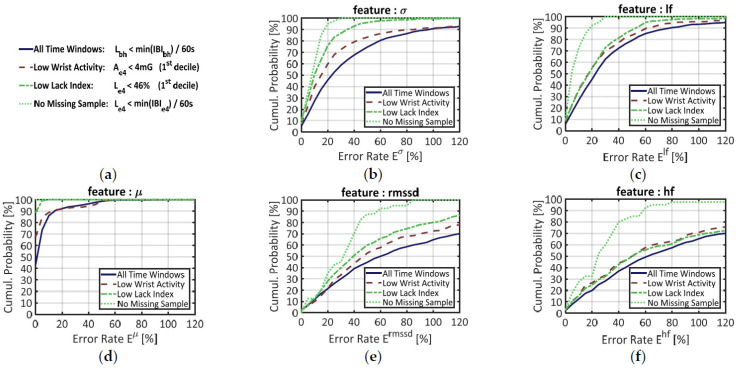
Cumulative probability of error for five features extracted from signal IBI_e4_, with signal IBI_bh_ as benchmark. (**a**) Main legend box; (**b**) Error on the standard deviation of the IBI (σ); (**c**) Error on the normalized power in the low-frequency range (lf); (**d**) Error on the mean IBI (µ); (**e**) Error in the root mean square of successive differences in the IBI (rmssd); (**f**) Error on the normalized power in the high-frequency range (lf). Each feature was computed on four different datasets: (i) over all time windows (blue plain line); (ii) over time windows that belong to the 1st decile of wrist activity A_e4_ (red dashed line); (iii) over time windows that belong to the 1st decile in the proportion of missing samples, estimated by the Lack index L_e4_ (green dashed-dotted line); (iv) over time windows showing no missing sample according to the Lack Index L_e4_ (green dotted line).

**Table 1 sensors-21-05357-t001:** Proportion of selected 60 s time windows over the whole dataset (upper line), and maximum activity level in chest (A_bh_) and wrist (A_e4_) in the selected time windows (lower lines). Over the whole dataset (i.e., without window selection), the maximum level for A_bh_ was 382 mG and the maximum level for A_e4_ was 825 mG. Column N° 2 corresponds to window selection from BioHarness SQI: C_hr_ (Equation (7)). Columns N° 3, 4, 5 correspond to window selection from the Lack Index of each IBI signal: L_bh_, L_pt_ and L_e4_ (Equation (6)).

Criterion	C_hr_	L_bh_	L_pt_	L_e4_
Selected windows	51.4%	48.2%	43.3%	0.6%
Maximum A_bh_	137 mG	137 mG	137 mG	7.05 mG
Maximum A_e4_	415 mG	415 mG	415 mG	19.1 mG

## Data Availability

The data reported in this study is not currently available in a public database.

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
