# Peer review of "Real-Time Quality Index to Control Data Loss in Real-Life Cardiac Monitoring Applications"

_sensors, 2021, doi:10.3390/s21165357_

Round 1
Reviewer 1 Report
This manuscript proposes a new method to assess the quality of the estimation of heart rate and heart rate variability (HRV) measurements from a photoplethysmography (PPG) recording and an electrocardiogram (ECG) recording obtained with two commercial devices. This work is original, relevant, and of interest to readers of the journal Sensors. The manuscript is well written. The methods and results are clearly described. There are only minor issues as described below.
- Abstract, lines 22 to 24. Please add some quantitative information in the results regarding the accuracy to complement the adjectives “high accuracy, poor accuracy and moderate accuracy”.
- Page 2, line 62. PPG is a standard measure of blood volume change, that is used to estimate the oxygen saturation in the blood (the PPG signal is not a direct measure of the oxygen saturation). I recommend clarifying this in the sentence.
- Page 6, section 2.2.1. The variation criterion is similar (but not the same) to the adaptive filtering developed by Niels Wessel et al. (doi:10.1007/s003990070035). This method is very effective to identify artifacts and heartbeats that are not from sinus origin. What are the differences and advantages of the new proposal? Please comment on the Discussion section.
- Page 6, line 285. How many neighbors around the mean were included to calculate the variation criterion? Since a 5-s moving average was computed, I assume the number of neighbors would vary with the mean heart rate. Could you provide data about the range of the number of neighbors that were used to calculate the variation criterion?
Author Response
Thank you for reviewing this article. Here is a point-by-point response to your comments.
1. Abstract, lines 22 to 24. Please add some quantitative information in the results regarding the accuracy to complement the adjectives “high accuracy, poor accuracy and moderate accuracy”.
This has been done for the three kinds of features.
2. Page 2, line 62. PPG is a standard measure of blood volume change, that is used to estimate the oxygen saturation in the blood (the PPG signal is not a direct measure of the oxygen saturation). I recommend clarifying this in the sentence.
Thank you for this precision.
3. Page 6, section 2.2.1. The variation criterion is similar (but not the same) to the adaptive filtering developed by Niels Wessel et al. (doi:10.1007/s003990070035). This method is very effective to identify artifacts and heartbeats that are not from sinus origin. What are the differences and advantages of the new proposal? Please comment on the Discussion section.
For the variation criterion, we followed the guidelines of Kemper et al, 2007 (https://doi.org/10.1203/PDR.0b013e318123fbcc), and Karlsson et al, 2012 (https://doi.org/10.1186/1475-925X-11-2), who showed this 30% variation limit from moving average preserved the short-term HRV features when few samples were deemed as abnormal.
This preprocessing algorithm is less sophisticated than the one proposed by Wessel et al (2000), which includes a parameter of dispersion from a weighted average of the surrounding beats. We did not include a dispersion parameter in our variation criterion, for two main reasons: (i) we expected the heart rate to evolve fast during physical activity ; (ii) we locally expected high missing sample rates (i.e., small sample sizes), which could affect the computation of a dispersion parameter.
This is only a guess: we did not back these hypotheses with experimental data. Besides, our preprocessing algorithm is quite perfectible since it relies on a short moving average (5s), making it unable to identify a too long succession of abnormal IBIs. Therefore, future work should include together different window sizes and variation criteria, like the adaptive filtering proposed by Wessel et al (2000). I added a discussion comment on the matter in the manuscript.
Although it has influenced our results, however, the outlier rejection procedure is not central to the method proposed in the current study: any efficient approach would fit and help estimating the quality of the heart rate signal. From my point of view, comparing our algorithm with another one in detail would blur the takeaway message of this article, which is: In real-life recordings, the Lack Index can be used to select time windows that are suitable to conduct a validation study or to reduce the estimation error on cardiac features. Following your comments (and those of another reviewer), the Discussion section was revised to underline this main contribution.
4. Page 6, line 285. How many neighbors around the mean were included to calculate the variation criterion? Since a 5-s moving average was computed, I assume the number of neighbors would vary with the mean heart rate. Could you provide data about the range of the number of neighbors that were used to calculate the variation criterion?
We chose a 5-second (rather than a 5-sample window for example) to make the algorithm robust to high missing sample rates. This parameter choice indeed varies the number of available samples to compute the moving average. This number depends on the local heart rate, but also on the number of missing samples at the current iteration (given that outlier samples were removed in a recursive fashion).
Following this remark, I computed the number of neighbors for each sample in the database after the last iteration in the process. For signal IBI_bh3 (sensor BioHarness), the mean number was 7.1, varying from 1 to 16. For signal IBI_e4 (sensor Empatica E4), the mean number was 5.0, varying from 1 to 15. These results were summarized in the manuscript.
Reviewer 2 Report
The paper is well written and it is focused on an interesting topic. The approach proposed is interesting. However, some points have to be improved:
Point 1. The introduction is well written and easy to follow. However, it is not mentioned some strategies used by researcher to evaluate missing values effect on HR and HRV. Please read also:
- Rossi A, Pedreschi D, Clifton DA, Morelli D (2020). Error Estimation of Ultra-Short Heart Rate Variability Parameters: Effect of Missing Data Caused by Motion Artifacts. Sensors, 20, 7122; doi:10.3390/s20247122
- Morelli D, Rossi A, Cairo M, Clifton DA (2019). Analysis of the Impact of Interpolation Methods of Missing RR-intervals Caused by Motion Artifacts on HRV Features Estimations. SENSORS, vol. 19, ISSN: 1424-8220, doi: 3390/s19143163
- Morelli D, Rossi A, Bartoloni L, Cairo M, Clifton AC (2021). SDNN24 Estimation from Semi-Continuous HR Measures. Sensors, 21, 1463; doi.org/10.3390/s21041463
Point 2 (line 177). Please provide more details about the three participants (e.g., height, weight, and body mass index)
Point 3 (line 188). Please provide the approval number of the ethical committee.
Point 4 (lines 221-228). Why do the participants wore the device also during day? Wrist worn devices are usually wore also during night.
Point 5 (lines 249-558). Are you sure about 0.215 seconds for the maximal limit? Moreover, what about the minimal value? Additionally, why do you chose 32 seconds as the limit for sync? This part is not clear. Please rephrase this paragraph.
Point 6 (lines 270-273). Why do you not apply interpolation approach for missing values as suggested in previous studies? For example, see: Morelli D, Rossi A, Cairo M, Clifton DA (2019). Analysis of the Impact of Interpolation Methods of Missing RR-intervals Caused by Motion Artifacts on HRV Features Estimations. SENSORS, vol. 19, ISSN: 1424-8220, doi: 10.3390/s19143163
Point 7 (line 322). The authors asserted that by “dividing μ by the sum of IBIs” you could obtain the number of beats (N). However, by logic and in accordance with the equation 2 this ratio is inverse. Please revise the text. I think the right phrase is “…dividing the sum of IBIs by their mean (μ)…”.
Point 8 (line 409). HRV parameters are usually computer on 5 minutes time window. Why do you chose to compute it on 1 minutes? How do you assess the accuracy of the ultrashort HRV values? Several previous study investigate the effect of ultrashort HRV with different accuracy in accordance with the HRV parameters. For example, 10 and 30 s recordings are the minimum time required to obtain accurate measures of RMSSD and SDNN, respectively. However, the shorter the time window is the higher is the bias for LF and HF.
Author Response
Thank you for the careful reading. Here is a point-by-point response to your comments.
Point 1. The introduction is well written and easy to follow. However, it is not mentioned some strategies used by researcher to evaluate missing values effect on HR and HRV. [...]
Thank you for sharing these references, especially the first one (Rossi et al, 2020) which is quite related to this article and was not yet published when the introduction to this article was written. I reported it in the introduction and discussion. The reasons why interpolation methods and long HRV features were not mentioned in the article are also detailed later in this reply.
Point 2 (line 177). Please provide more details about the three participants (e.g., height, weight, and body mass index)
Unfortunately, we did not collect this kind of information. This database was initially designed for the detection of acute stress events on healthy subjects, and heart activity was not the only physiological signal of interest.
Point 3 (line 188). Please provide the approval number of the ethical committee.
It is now reported in the same paragraph.
Point 4 (lines 221-228). Why do the participants wore the device also during day? Wrist worn devices are usually wore also during night.
We were only interested in daily activities because this study was initially designed for stress detection. Additionnally, the sensors had to be recharged during night because the experiment had to last over an entire week.
Point 5 (lines 249-558). Are you sure about 0.215 seconds for the maximal limit? Moreover, what about the minimal value? Additionally, why do you chose 32 seconds as the limit for sync? This part is not clear. Please rephrase this paragraph.
2^15 milliseconds make approximatively 32 seconds. This maximum limit is imposed by sensor BioHarness, which probably stores its IBI signal in a 16-bit variable. The adverse effect is a memory overflow : as soon as no heartbeat is detected during more that 32 seconds, the counter restarts from 0 and the IBI signal goes out of sync with the other signals. This explanation is just a guess (we did not receive it from the Zephyr company), so we did not write it explicitly in the submitted manuscript. This has been cautiously done following your comment.
Point 6 (lines 270-273). Why do you not apply interpolation approach for missing values as suggested in previous studies? [...]
In this study, our main point was to use the lack of samples to measure the quality of the IBI signal. Interpolation approaches could be useful only while extracting features from the PPG-based IBI signal (i.e., IBI_e4) to compare them with the ECG-based IBI signal (IBI_bh). However, no interpolation method could reliably compensate the high missing sample rates found for signal IBI_e4 in our database (above 50% for 9 time windows out of 10, see Figure 3).
I still added a comment on interpolation approaches in the discussion.
Point 7 (line 322). The authors asserted that by “dividing μ by the sum of IBIs” you could obtain the number of beats (N). However, by logic and in accordance with the equation 2 this ratio is inverse. Please revise the text. I think the right phrase is “…dividing the sum of IBIs by their mean (μ)…”.
Thank you for this remark. The mistake has been corrected.
Point 8 (line 409). HRV parameters are usually computer on 5 minutes time window. Why do you chose to compute it on 1 minutes? How do you assess the accuracy of the ultrashort HRV values? Several previous study investigate the effect of ultrashort HRV with different accuracy in accordance with the HRV parameters. For example, 10 and 30 s recordings are the minimum time required to obtain accurate measures of RMSSD and SDNN, respectively. However, the shorter the time window is the higher is the bias for LF and HF.
These comments are quite relevant. At the beginning, the current study also considered several window sizes W for each analysis (i.e.: W = 30 seconds, W = 1 minute and W = 5 minutes). This variation in W was finally withdrawn from the article because it made the Results section too complex. It blurred our takeaway message, which is: In real-life recordings, the Lack Index L can be used to conduct a validation study or to select time windows that are suitable for feature extraction.
Subsequently, the reason why 1-minute time windows were preferred to 5-minutes time windows was mainly statistical. To compare HRV features extracted from PPG with a "gold standard", we had to select time windows where the ECG-based IBI signal showed no missing sample. In this process, setting W = 5min yielded a much smaller dataset (259 values for each feature, covering ∼20% of the initial database) than setting W = 1min (3370 values, covering ∼50% of the initial database: see Table 1).
The choice of ultra-short HRV features to validate sensor Empatica E4 was thus driven by the availability of benchmark values in our database. Still, the combined influence of parameters W (window size) and L (missing sample rate) on each feature's error rates is obviously a subject for future work on real-life cardiac data. This would make a nice extension of the article you proposed in Point 1 (Rossi et al., 2020). In that perspective, the "suitable" level of error for each HRV feature may depend on the desired application (e.g., stress detection or heart event prevention).
A comment was made on that matter in the discussion.
Round 2
Reviewer 2 Report
The authors fully answered all of my doubts. The paper is now acceptable for publication. Congrats for this interesting work!